# Mutations Linked to Insecticide Resistance Not Detected in the *Ace-1* or *VGSC* Genes in *Nyssorhynchus darlingi* from Multiple Localities in Amazonian Brazil and Peru

**DOI:** 10.3390/genes14101892

**Published:** 2023-09-29

**Authors:** Sara A. Bickersmith, John D. Jurczynski, Maria Anice Mureb Sallum, Leonardo S. M. Chaves, Eduardo S. Bergo, Gloria A. D. Rodriguez, Clara A. Morante, Carlos T. Rios, Marlon P. Saavedra, Freddy Alava, Dionicia Gamboa, Joseph M. Vinetz, Jan E. Conn

**Affiliations:** 1Wadsworth Center, New York State Department of Health, Albany, NY 12201, USA; sara.bickersmith@health.ny.gov (S.A.B.); john.jurczynski@health.ny.gov (J.D.J.); 2Department of Biomedical Sciences, School of Public Health, University at Albany, State University of New York, Albany, NY 12222, USA; 3Departamento de Epidemiologia, Faculdade de Saúde Pública, Universidade de São Paulo, São Paulo 01246-904, Brazil; masallum@usp.br (M.A.M.S.); leonardosuveges@usp.br (L.S.M.C.); 4Secretaria de Estado da Saúde de São Paulo, Instituto Pasteur, São Paulo 01027-000, Brazil; edusteber@uol.com.br; 5Laboratorio de Referencia Regional de Loreto, Gerencia Regional de Salud de Loreto (GERESA), Loreto 16001, Peru; gloriaadiaz@yahoo.com (G.A.D.R.); cdelaguilamorante@gmail.com (C.A.M.); ctong32@gmail.com (C.T.R.); 6Laboratorio ICEMR-Amazonia, Laboratorios de Investigacion y Desarrollo, Facultad de Ciencias e Ingeniería, Universidad Peruana Cayetano Heredia, Lima 15102, Peru; msaavedraromero75@gmail.com (M.P.S.); dionicia.gamboa@upch.pe (D.G.); joseph.vinetz@yale.edu (J.M.V.); 7Gerencia Regional de Salud de Loreto (GERESA), Loreto 16001, Peru; ffalare@hotmail.com; 8Instituto de Medicina Tropical “Alexander von Humboldt”, Universidad Peruana Cayetano Heredia, Lima 15102, Peru; 9Departamento de Ciencias Celulares y Moleculares, Facultad de Ciencias e Ingeniería, Universidad Peruana Cayetano Heredia, Lima 15102, Peru; 10Section of Infectious Diseases, Department of Internal Medicine, Yale University School of Medicine, New Haven, CT 06520, USA

**Keywords:** *Nyssorhynchus darlingi*, insecticide resistance, *Ace-1*, *VGSC*, Brazil, Peru

## Abstract

Indoor residual spray (IRS), mainly employing pyrethroid insecticides, is the most common intervention for preventing malaria transmission in many regions of Latin America; the use of long-lasting insecticidal nets (LLINs) has been more limited. Knockdown resistance (*kdr*) is a well-characterized target-site resistance mechanism associated with pyrethroid and DDT resistance. Most mutations detected in acetylcholinesterase-1 (*Ace-1*) and voltage-gated sodium channel (*VGSC*) genes are non-synonymous, resulting in a change in amino acid, leading to the non-binding of the insecticide. In the present study, we analyzed target-site resistance in *Nyssorhynchus darlingi*, the primary malaria vector in the Amazon, in multiple malaria endemic localities. We screened 988 wild-caught specimens of *Ny. darlingi* from three localities in Amazonian Peru and four in Amazonian Brazil. Collections were conducted between 2014 and 2021. The criteria were Amazonian localities with a recent history as malaria hotspots, primary transmission by *Ny. darlingi*, and the use of both IRS and LLINs as interventions. Fragments of *Ace-1* (456 bp) and *VGSC* (228 bp) were amplified, sequenced, and aligned with *Ny. darlingi* sequences available in GenBank. We detected only synonymous mutations in the frequently reported *Ace-1* codon 280 known to confer resistance to organophosphates and carbamates, but detected three non-synonymous mutations in other regions of the gene. Similarly, no mutations linked to insecticide resistance were detected in the frequently reported codon (995) at the S6 segment of domain II of *VGSC*. The lack of genotypic detection of insecticide resistance mutations by sequencing the *Ace-1* and *VGSC* genes from multiple *Ny. darlingi* populations in Brazil and Peru could be associated with low-intensity resistance, or possibly the main resistance mechanism is metabolic.

## 1. Introduction

Indoor residual spray (IRS) and, more recently, long-lasting insecticidal nets (LLINs), have been used as the primary malaria vector control interventions due to their cost-effectiveness and protection against vectors that feed and rest mainly indoors [1,2]. In contrast, in Latin America, many vectors feed and rest primarily outdoors (exophagy and exophily, respectively). This behavior, together with the widespread existence of houses with incomplete walls, communities where unprotected travel-related occupations are common [3,4,5], and the temporally irregular application of IRS in many communities diminishes the effectiveness of insecticide-based interventions throughout Latin America in relation to some other endemic malaria regions [6,7].

In 2007, the World Health Organization’s (WHO) recommendations for LLIN distribution were broadened to include all individuals in endemic malaria areas, rather than solely pregnant women and children under the age of 5 [8]. Simultaneously, the WHO guidelines called for endemic malaria nations to adopt triennial LLIN mass distribution campaigns, which would allocate one LLIN for every two people per household in a given region, aspiring to universal coverage [9]. Following these updates, mass distribution campaigns became more prevalent in Latin America, particularly in the Amazon region. In Peru, the Project for Malaria Control in Andean Border Areas (PAMAFRO) was mainly responsible for a 70% decrease in cases from 2005 to 2011 [9]. Since the program was discontinued, malaria cases have increased fairly steadily due to a decline in international and domestic funding [10]. The sharpest increase in cases throughout the Latin American region occurred between 2014 and 2017, when case incidence nearly doubled [11]; ironically, incidence has apparently decreased with the recent COVID-19 pandemic [12]. An encouraging sign is that the Ministry Health, Peru, adopted a new plan as of January 2022 to eliminate malaria by 2030 [13].

Between 2010 and 2019, an estimated 28 endemic malaria countries (of 82) detected insecticide resistance to all four of the most commonly used insecticides (pyrethroids, organochlorines, carbamates, and organophosphates), and nearly all (73/82) have reported resistance to at least one class [14,15]. Worldwide, the exposure of mosquitoes to any of these classes of insecticides [16], whether for public health or agricultural use, has the potential to be a strong selective force that favors the survival of resistant populations [17]. In Latin America, there is relatively little entomological surveillance and few published studies on insecticide resistance (IR), except for *Nyssorhynchus albimanus* [18]. The primary malaria vector in the Amazon, *Ny. darlingi*, is generally susceptible to insecticides [17,19] and exhibits exo- and endophagic behavior, depending on its local circumstances [20]. Nevertheless, resistance based on bioassays using WHO paper bioassays [21] or CDC bottle bioassays [22,23] has been reported for *Ny. darlingi* to pyrethroids and carbamates (Bolivia), pyrethroids (Brazil), pyrethroids and organochlorines (Colombia), and pyrethroids, carbamates, and organophosphates (Peru) [24].

IR in mosquitoes can result from several mechanisms. One of the best-characterized is knockdown resistance (*kdr*) or target-site point mutations, commonly associated with pyrethroid and DDT resistance [25,26]. The site of most *kd*r mutations is either the voltage-gated channel (*VGSC*) corresponding to the *VGSC* gene, located in the transmembrane segment IIS6 or the linker regions that connect domains III and IV, or the acetylcholinesterase-1 (*Ace-1*) gene in anophelinae and other mosquito species. The Asian malaria vector *Anopheles culicifacies* was one of the earliest anophelines to be found to be resistant to DDT in Gujerat State, India, using DDT-impregnated paper [27]. On the other hand, the first instance of DDT resistance in *Ny. darlingi* was not detected until 1990, using WHO techniques for susceptibility [28] in the Choco region of northwestern Colombia [29]. The second mechanism, behavioral modification, is a change in behavior, such as the avoidance of or repellence in response to insecticide-impregnated surfaces, or a modification in location (outdoors vs. indoors; dispersion to untreated houses), or time of day of blood seeking. This has been called *qualitative behavioral resistance* recently [30]. The first systematic study of *Ny. darlingi* behavioral modification was a demonstration of repellency to DDT-impregnated house surfaces in Amazonian Brazil [31]. A second example is the detection of fewer *Ny. darlingi* biting outdoors and more indoors between the distributions of LLINs (as the nets aged and became non-repellent) in Amazonian Peru [32]. Another mechanism is increased metabolism (detoxification/sequestration) through mixed-function oxidases (MFO) and non-specific esterases (NSE) that have developed during insect evolutionary history as protection against a range of plant toxins [33,34].

Of these mechanisms, only target-site—considered to be the most accurate indicator of resistance—can be identified using molecular assays [35]. These assays detect amino acid substitutions that give rise to non-synonymous amino acid changes in insecticide targets, ultimately preventing the insecticide from binding, leading to resistance [36]. Among the genes of interest are voltage-gated sodium channel (*VGSC*) and acetylcholinesterase-1 (*Ace-1*), which encode for the target binding of pyrethroids/organochlorines and carbamates/organophosphates, respectively [26,36]. Mutations linked most frequently with IR are L995F and L995S in *VGSC* [37] and G280S in *Ace-1* [38]. Other mutations include L995C, L995W, V991L, and V994S in *VGSC* [38], and, in *Ace-1*, A221T and S216T [18]. Codon 995 in *VGSC* and codon 280 in *Ace-1* were referred to formerly as 1014 and 119, respectively [18]. Despite the accuracy of molecular assays, susceptibility (or phenotypic) bioassays are advantageous because hundreds of mosquitoes can be tested simultaneously with relatively simple equipment [24,35]. Having data from both molecular assays and phenotypic bioassays is ideal for assessing resistance frequency [36].

The combination of resurgent cases of malaria since 2014 and the widespread increase in the use of insecticides, yet the scarce reporting of resistant vectors, suggest a potential knowledge gap in IR throughout the Latin American region. IR detection requires the strategic selection of localities in which mosquito samples are collected and analyzed. Commonly used selection criteria from previous reports include accessibility by land or water, a history of insecticide use, a high malaria prevalence, and a sample size sufficient even for a population with a low frequency of resistant vectors [39,40]. This study aims to help close the knowledge gap in *Ny. darlingi* by using molecular assays for a relatively large sample size covering multiple localities, uncovering novel codon mutations, and providing insight that may assist in improving malaria vector control interventions.

## 2. Materials and Methods

### 2.1. Sample Collection

Samples of *Ny. darlingi* were collected in several rural and riverine localities throughout Amazonian Brazil and Peru in 2014–2021 (Table 1; Figure 1). Sample locations were based on information from local health officials about exposure to IRS or LLINs. In Cruzeiro do Sul, Acre, Brazil, local public health personnel (Brazil Ministry of Health and Instituto Evandro Chagas) demonstrated phenotypic resistance via discriminating concentration bioassays to deltamethrin and cypermethrin from 2012 to 2014 [41]. In the villages of Gamitanacocha and Zungarococha, Peru, local health authorities (Laboratorio Referencial de Salud Pública Loreto) demonstrated phenotypic resistance and possible resistance, respectively, to pyrethroids in 2018 [41]. Furthermore, Zungarococha and Cahuide are both along a highway where extensive IRS was used during a major malaria outbreak in 2012 [10].

Mosquito collections were performed indoors, peridomestically, or in forest edge settings during various time periods throughout the evening using human landing catch, barrier screen sampling, or Shannon traps [32,42,43]. Details on the forest cover level [42], field collection protocols [32,43], and *Plasmodium vivax* malaria incidence [43] have been reported previously. Collected mosquitoes were identified to species morphologically by trained personnel using regional taxonomic keys [44,45,46] and stored on silica gel until a genetic analysis was conducted. Genomic DNA was extracted from whole mosquitoes using the Qiagen DNeasy Blood & Tissue Kit (Qiagen, Germantown, MD, USA).

### 2.2. Molecular Analysis

Samples used for the genetic analysis were selected randomly from the available specimens in storage using a random number generator [47]. As year-round samples from Cahuide were available, we selected specimens from both the rainy and dry seasons [32]. The amplification of a 228 bp fragment of the *kdr* region of the *VGSC* gene, between exons 20 and 21, was performed in a 20 μL PCR mixture containing a 1.0–15 ng/μL DNA template, 1× AllTaq Master Mix (Qiagen), and 0.5 μM of each primer: AAKDRF2 and AAKDRR2, with the same cycling conditions as in [16]. Amplification of a 456 bp fragment of Exon 2 in the *Ace-1* gene was performed in a 25 μL PCR mixture containing a 1.0–15 ng/μL DNA template [43], 2 U of *Taq* DNA Polymerase (Qiagen), 10× PCR Buffer (Qiagen) containing 1.5 μM of MgCl_2_, a supplemental 0.5 μM of MgCl_2_ totaling 2.0 μM of MgCl_2_, 0.2 μM of dNTPs, and 0.4 μM of each primer ACE1DAF and ACE1DAR, with the same cycling conditions as in [18].

SSamples were Sanger sequenced in forward and reverse directions at the Wadsworth Center Advanced Genomic Technologies Core (New York State Department of Health). Chromatograms of each sample were cleaned, converted into consensus sequences, translated, and exported to FASTA files using Geneious Prime Version 2020.2 [48]. Consensus sequences were aligned using ClustalW in MegaX Version 10.1.7, then analyzed in comparison to existing *VGSC* and *Ace-1* sequences of *Ny. darlingi* from GenBank [16]. Unique sequences for *VGSC* were deposited in GenBank under the accession numbers: OR260704–OR260712 and *Ace-1* under the accession numbers: OR260713–OR260857. The sSequences for both genes were examined for non-synonymous (amino acid change) and synonymous (no amino acid change) point mutations, with special focus on the documented codons known to convey insecticide resistance within these gene regions.

## 3. Results

### 3.1. Detection of Voltage-Gated Sodium Channel (VGSC) Mutations

A total of 495 wild-caught *Ny. darlingi*, 296 from Peru and 199 from Brazil, were successfully sequenced for the *kdr* target-site resistance region of the *VGSC* gene (Table 1). SSequences were aligned with the aid of the *Ny. darlingi VGSC* sequences available in Genbank [16]. We detected nine unique genotypes (denoted as V1–V9, corresponding to Genbank IDs OR260704–OR260712), with only the susceptible genotype TTA (leucine) observed at codon position 995 (Figure 2). The only point mutations detected were in the intron and primer regions.

### 3.2. Detection of Acetylcholinesterase-1 (Ace-1) Mutations

We successfully sequenced 493 *Ny. darlingi*, 295 from Peru and 198 from Brazil, for the *Ace-1* gene (Table 1), and detected 145 unique genotypes (named AC1–AC145, corresponding to Genbank IDs OR260713–OR260857). The susceptible genotype GGG/GGG or GGG/GGC (glycine) was observed at codon 280 across all samples. Three non-synonymous mutations were detected in samples from three populations (Humaitá and Mâncio Lima, Brazil; and Cahuide, Peru) within other regions of the *Ace-1* gene not known to convey insecticide resistance (Figure 3).

### 3.3. Data Management

Unique sequences of *Ny. darlingi* for *VGSC* were deposited in GenBank under accession numbers: OR260704–OR260712, and for *Ace-1,* under accession numbers: OR260713–OR260857. All sample results will be deposited in VectorBase pending publication.

## 4. Discussion

A molecular analysis of a ~228 bp fragment that encodes for the *kdr* target-site resistance region of the *VGSC* gene and a ~456 bp fragment of the *Ace*-1 gene did not detect any non-synonymous mutations in the specimens of *Ny. darlingi* from endemic malaria areas of Brazil and Peru in the current study, similar to results of a recent analysis of *Ny. darlingi* from locations in Brazil (Manaus, Unini River, Jau River in Amazonas State, and Porto Velho, Rondônia state) and Colombia (Tagachi and Chocó Department) [49]. An investigation of specimens of *Ny. darlingi* from Chocó Department, Colombia, that had been demonstrated to be phenotypically susceptible and resistant, sequenced for the same region of the *VGSC* gene, also did not reveal any *kdr* mutations [16]. However, the classic L1014F *kdr* mutation has been detected in other important anopheline malaria vectors, i.e., *Nyssorhynchus albimanus* [16,50,51] and *Nyssorhynchus albitarsis* s.s. [52]. Other species of Latin American malaria vectors evaluated with these molecular assays include *Anopheles vestitipennis* and *Anopheles pseudopunctipennis*, both of which exhibited genotypic susceptibility [16]. The lack of genotypic evidence of IR in *Ny. darlingi* could be a reflection of limited regional data, rather than the absence of resistance [53]. On the other hand, Floch [54] suggested that frequent reintroduction of wild susceptible populations of *Ny. darlingi* from forest into village populations could reduce selection for insecticide resistance. This hypothesis received some support from a study in the Porto Velho area in Rondônia state, Brazil, that detected seasonal gene flow between forested and urban populations of *Ny. darlingi* [55].

Several previous studies of *Anopheles* malaria vectors have attributed a proportion of recent malaria case resurgence to increased outdoor biting and insecticide resistance (IR) following mass distribution campaigns [56,57,58,59], although there is no evidence in support of this latter trend for *Ny. darlingi* across Latin America (the scale-up of LLIN distribution has been limited compared with Africa) or after the completion of the intensive PAMAFRO project in Peru. Even though daytime biting behavior in members of the *An. gambiae* complex has been hypothesized to increase transmission [60], in Latin America, there has been scant investigation into this phenomenon, except for observations of *Ny. darlingi* biting during the day in forested French Guiana malaria hotspots [61].

The hot, rainy climate of the Amazon basin is optimal for mosquito habitats [62]; however, anthropogenic landscape changes—namely, forest fragmentation and an increased ecotone density—suggest that vector behavior (mainly exophagy and exophily in *Ny. darlingi*) and distribution (i.e., along ecotones for *Ny. darlingi*) in one location may not be generalizable to an entire region [63,64]. For example, during a recent malaria outbreak in French Guiana, *Ny*. *darlingi* was the only anopheline collected both outdoors and indoors, and its abundance was exceptionally high, possibly attributed to regional deforestation, and/or the higher than average rainy and dry seasons in 2017 [65]. *Ny. darlingi* has also been collected biting during the day in French Guiana forests [66,67] and along the Maroni River, a former malaria hotspot, in Suriname [68]. This appears to be a focal phenomenon in *Ny. darlingi*, perhaps a behavioral avoidance response to IRS or LLINs.

The majority of IR reports in Latin America are based on bioassay data from the Amazon Basin or Central America [24,36]. However, the recent genotypic reporting of several Brazilian samples of *Ny. albitarsis s.s.* showed heterozygous L995F mutations in *VGSC* [52], and a sample of Guatemalan *Ny. albimanus* had a heterozygous G280S mutation in *Ace-1* [69]. Both of these reports inferred that agricultural insecticide use was the driver of IR, and a recent review of the contribution of agricultural insecticides and increasing insecticide resistance in malaria vectors found a strong association across Africa that could be affected by crop type (especially cotton and vegetables), urban development, and the strategies undertaken for farm pest management [70]. Questionnaires and insecticide susceptibility bioassays utilized in a field study in two South Côte d’Ivoire communities determined that local mosquito vectors were resistant to three of four insecticides tested, and the authors highlighted the need for collaboration between the public health and agricultural sectors to develop interventions that would benefit both [71]. Resistance in the important regional malaria vector, *Ny. albimanus*, has been detected in Central America, Panama, and northwestern coastal Peru, linked mainly to agriculture in general and rice cultivation in coastal Peru in particular [17].

Public health insecticide use can exert comparable selective pressure on malaria vectors [72], including, for example, the organophosphate malathion used in Brazilian public health for the arboviral vectors *Aedes aegypti* and *Aedes albopictus* to reduce the transmission of viruses such as dengue, chikungunya, and Zika [73]. Resistance in the vector *Culex quinquefasciatus* in Brazil has been detected for organophosphates, carbamate, DDT, pyrethroids, and biolarvicides; the concern for such resistance to arise in malaria vectors in Brazil, where they co-occur with *Cx. quinquefasciatus*, is limited to Fortaleza, Ceará state, and parts of Mato Grosso state [74].

For control of adult mosquitoes, IRS on interior house walls will kill resting mosquitoes; some also repel mosquitoes such that they modify their behavior and rest outdoors [17]. Based on an evaluation of the residual effects of four insecticides (deltamethrin, pyrethroids, lambda-cyhalothrin, and etofenprox) used on a range of wall materials in Amazonian Brazil [75], the Brazilian National Malaria Control Plan has been consistently using etofenprox PM 20% for residual spray in houses since 2013 [19], although a similar study of six insecticides in Amapá, Brazil, by Correa et al. [76] found that deltamethrin WG at 0.025 gm/m^2^ had the highest residual effects. In Peru, pyrethroid deltamethrin 5% is the most commonly applied insecticide for IRS [32].

Other approaches to tackling insecticide resistance consider biological control in general [77] or the replacement of synthetic compounds with plant-based compounds formulated as bioinsecticides, reviewed in Demirak and Canpolat [15]. The classes of compounds described and discussed were phytochemicals, pheromones, microbial pesticides, and plant-incorporated protectants; selected candidate compounds demonstrate larvicidal, adulticidal, and repellent properties. In general, their advantages are, compared to synthetic compounds, a lower toxicity, target specificity, being highly effectivity in small quantities, and they are biodegradable. Despite considerable promise, these products remain in various stages of development, and have not yet been field-tested for use against malaria vectors. As many target insects have evolved successful resistance mechanisms to most classes of insecticides, the evolution of different modes of action against plant-based insecticides could temper the early enthusiasm for such novel products [33].

## Figures and Tables

**Figure 1 genes-14-01892-f001:**
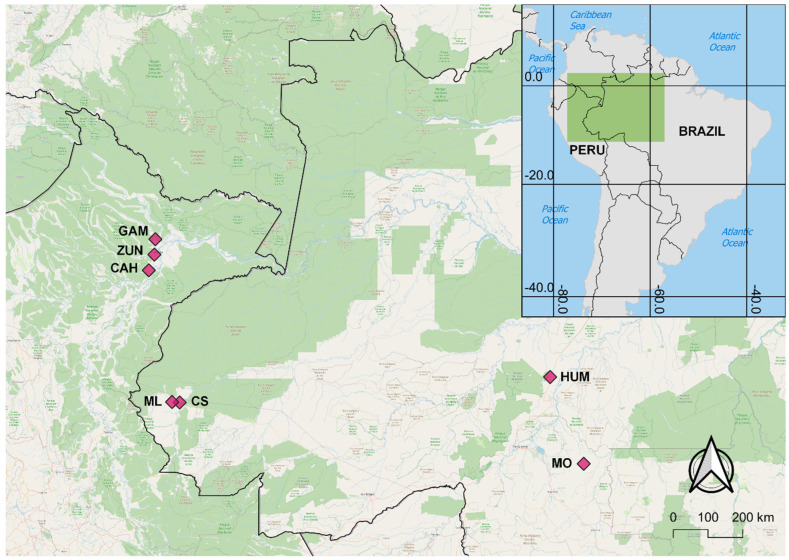
Map of *Ny. darlingi* collection sites in Amazonian Peru (GAM: Gamitanacocha; ZUN: Zungarococha; and CAH: Cahuide) and Brazil (ML: Mâncio Lima; CS: Cruzeiro do Sul; HUM: Humaitá; and MO: Machadinho d’Oeste).

**Figure 2 genes-14-01892-f002:**
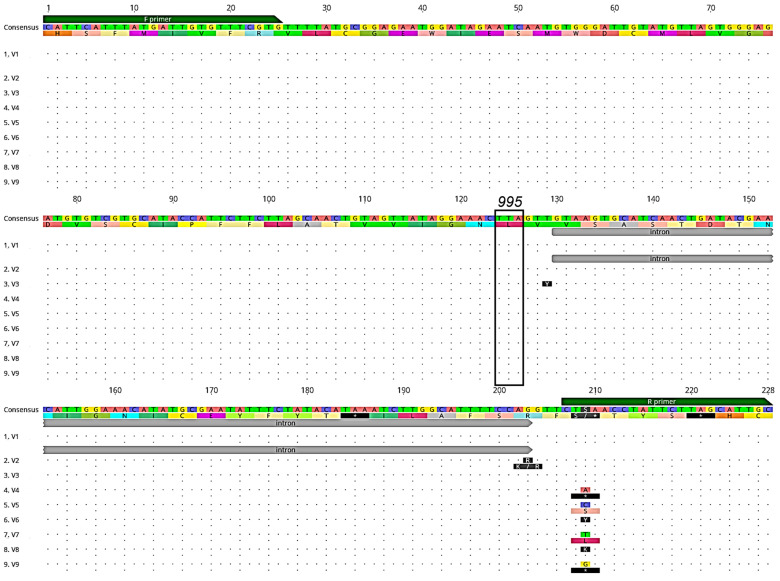
Alignment of DNA genotypes (V1-V9) and amino acid translations (below each DNA sequence) of the *VGSC* gene *kdr* region sequenced from *Ny. darlingi* collected in Peru and Brazil. Dots indicate no base or amino acid change from consensus sequence. Forward (F primer) and reverse primer (R primer) regions (green boxes), intron position (gray), and codon 995 (black box), associated with pyrethroid and DDT resistance are denoted (image obtained from Geneious version 2020.2 created by *Biomatters*).

**Figure 3 genes-14-01892-f003:**
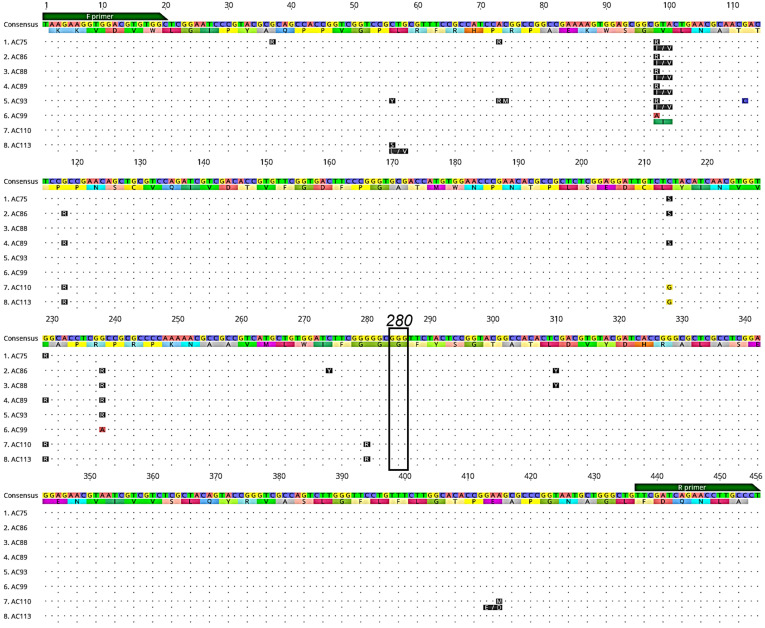
Alignment of a subset of *Ace-1* DNA genotypes and amino acid translations (below each DNA sequence) displaying non-synonymous mutations (blue lines) detected in *Ny. darlingi* samples sequenced from Peru and Brazil. Dots indicate no base or amino acid change from consensus sequence. Forward (F primer) and reverse primer (R primer) regions (green boxes) and codon 280 (black box), which is associated with organophosphate and carbamate resistance, are denoted (image obtained from Geneious version 2020.2 created by Biomatters).

**Table 1 genes-14-01892-t001:** Summary of *Ny. darlingi* samples sequenced for the *Ace-1* and *VGSC* genes.

Locality	State/Dept	Country	Latitude	Longitude	Collection Year	*Ace-1* N	*VGSC* N
Cruzeiro do Sul (CS)	Acre	Brazil	−7.631889	−72.688722	2015	49	50
Humaitá (HUM)	Amazonas	Brazil	−6.980687	−63.100031	2016	49	49
Machadinho d’Oeste (MO)	Rondônia	Brazil	−9.193528	−62.230107	2015	50	50
Mâncio Lima (ML)	Acre	Brazil	−7.620124	−72.885559	2015	50	50
Gamitanacocha (GAM)	Loreto	Peru	−3.426000	−73.318000	2018	50	50
Zungarococha (ZUN)	Loreto	Peru	−3.824560	−73.343880	2019; 2021	106	107
Cahuide (CAH)	Loreto	Peru	−4.230350	−73.487833	2014; 2019	139	139
					**TOTAL**	**493**	**495**

N: Number of individual mosquitoes sequenced.

## Data Availability

All sequences were deposited into GenBank (detailed above) and relevant mosquito data will be added to VectorBase post-publication.

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
