# Peer review of "Mutations Linked to Insecticide Resistance Not Detected in the Ace-1 or VGSC Genes in Nyssorhynchus darlingi from Multiple Localities in Amazonian Brazil and Peru"

_genes, 2023, doi:10.3390/genes14101892_

Round 1
Reviewer 1 Report
The article of Sara A. Bickersmith with coauthors is concern of epidemiologically important topic of insecticide resistance in malaria vector Nyssorhynchus darlingy. This mosquito is widely spread species in South American that represents significant threat of the malaria transmission in this region. The authors explored the target-site insecticide resistance in Ny. darlingi from multiples locations in Amazonian Peru and Brazil. Surprisingly, they detected only synonymous mutations in acetylcholinesterase-1 and voltage-gated sodium channel genes in positions that are usually associated with insecticide resistance in other species of mosquitoes. However, they found non-synonymous mutations in other positions of these genes. These results may suggest presence of a different mechanism of insecticide resistance in this species and thus this topic requires further investigation. I think that the results of this manuscript are novel and important and the manuscript is well written.
Below are some suggestions for the manuscript improvement:
1. I suggest using abbreviations more carefully and to be consistent through the text. In abstract the IR (insecticide resistance) is used without explanation (line 41). Then in the entire text IR abbreviation is not used but in the discussion section it is finally explained (line 215). Other abbreviations, human landing catch (HLC), barrier screen sampling (BSS), Shannon traps (ST) in line 130, and plant-incorporated protectants (PIPs) in line 271 are used only once and thus using such abbreviations is not necessary.
2. In line 86-88 author wrote: “Insecticide resistance in mosquitoes can be derived from several mechanisms including behavioral (plasticity) [25], metabolic (enzymes that metabolize (detoxify) or sequester insecticides) [26,27], and target-site (genotype) [28,29].” I think that the meaning of this sentence is hard to understand. I suggest describing each mechanism of insecticide resistance in more details.
3. I also suggest making letter indicating nucleotides in figures 2 and 3 brighter.
Reviewer 2 Report
In this manuscript, Dr. Bickersmith and colleagues report the evaluation of mutations linked to insecticide resistance by genotypic detection from multiple Ny. darlingi populations from Brazil and Peru. The authors sequenced a relatively large sample size from fragments of Ace-1 and VGSC, however mutations linked to insecticide resistance were not detected in any of the samples, indicating low-intensity resistance in the mosquitoes from the localities studied or a different resistance mechanism is present in that population.
This paper is well written. The introduction and discussion bring relevant publications, and the methods section is sufficiently described with an interesting discussion about the topic.
Find below suggestions/comments in the hope of improving the manuscript.
How Ny. darlingi was identified?
Line 132-133: I suggest moving this paragraph to section 2.2 Molecular Analysis.
Line 230: Abbreviate Nyssorhynchus
Figures:
Both figures are very small and almost impossible to read. It needs to be enlarged.
Is there any correlation between the unique genotypes found and the localities where they were collected?
